# Applicability of a Recreational-Grade Interferometric Sonar for the Bathymetric Survey and Monitoring of the Drava River

**Ákos Halmai** [1,*] **, Alexandra Gradwohl-Valkay** [2] **, Szabolcs Czigány** [1] **, Johanna Ficsor** [2] **,**
**Zoltán Árpád Liptay** [3] **, Kinga Kiss** [1] **, Dénes Lóczy** [1] **and Ervin Pirkhoffer** [1]

[1] University of Pécs, Faculty of Sciences, Institute of Geography and Earth Sciences, 7624 Pécs, Ifjúság útja 6., Hungary; sczigany@gamma.ttk.pte.hu (S.C.); kissk@gamma.ttk.pte.hu (K.K.); loczyd@gamma.ttk.pte.hu (D.L.); pirkhoff@gamma.ttk.pte.hu (E.P.)

[2] University of Pécs, Faculty of Sciences, Doctoral School of Earth Sciences, 7624 Pécs, Ifjúság útja 6., Hungary; valkays2@gamma.ttk.pte.hu (A.G.-V.); johannaficsor@hotmail.com (J.F.)

[3] General Directorate of Water Management, Hungarian Hydrological Forecasting Service, 1012 Budapest, Márvány u. 1/d., Hungary; liptay.zoltan@ovf.hu

*** Correspondence: halmaia@gamma.ttk.pte.hu; Tel.: +36-30-434-1648

**Abstract:** Sonar survey of shallow water bodies has challenged scientists for a long time. Although these water courses are small, still they have an increasing ecological, touristic and economical role. As maritime sonars are non-ideal tools for shallow waters, the bathymetric survey of these rivers has been taken with cross-sectional methods. Due to recent developments, interferometric surveying technology have also burst into the market of recreational-grade fish-finders. The objective of the current study was the development of a novel, complex and integrated surveying technique which is affordable, robust and applicable even at low water levels. A recreational-grade sonar system was assembled and mounted on a double-hull vessel and connected with a geodetic Global Navigation Satellite System (GNSS) device. We have developed a novel software which enables the bridging between a closed sonar file format and the commonly used Geographic Information System (GIS) datasets. As a result, the several month-long conventional bathymetric survey of the 146 km-long reach of the Drava River was reduced to 20 days and provided channel bathymetry of many orders of magnitude higher than the classical methods. Additionally, a large number of spatial derivatives were generated which enables the analysis of channel morphology, textural variation of channel sediments and the accurate delineation of navigational routes.

**Keywords:** remote sensing; active sonar; bathymetry; interferometry; recreational-grade sonar; fish finder; Drava River

---

## 1. Introduction

The field of sonar technology and underwater acoustics was primarily developed for military purposes [1,2]. All major geopolitical actors developed their own submarine and anti-submarine systems during the World Wars and the Cold War, and invested significant efforts into seafloor mapping [3].

The original idea of the sonar was an implementation of a single-beam device capable of simple depth measurements. However, until the late 1980s, these systems were superseded by newer sonar technologies, which were suitable to survey larger areas of the seafloor and to detect reflectance, material composition and sediment properties [3,4]. These applications are originated from naval research, but they have slowly emerged into the commercial environment; however, the primary

application of the sonar systems is still closely related to the seas. Earth sciences, in general, have also gained several discoveries through applied underwater acoustics, like the mapping of mid-ocean ridges and trenches, as well as the acquisition of topographic and tectonic data are due to sonar measurement [3]. Sonars have also been successfully applied for marine environment mapping, cultural heritage assessment and in general in the field of archeology [4,5]. Sonars—originally developed for seafloor surveys—were professional, high precision, high value and relatively large devices and their availability and applicability were limited in shallow and/or small-scale riverine environments. Small rivers, however present specific challenges for bathymetric surveys, as large devices cannot be mounted on shallow-draft vessels and traditional sonars are not designed to survive frequent impacts with large floating or non-visible subsurface objects. From the viewpoint of riverine traffic, small water courses have low priority; therefore, no significant financial resources are available for their survey.

Meanwhile, the sonar manufacturers discovered a new business segment to enhance their profit: the segment of recreation and recreational fishing and boating. They started to sell an already existing technology in a minified and simplified user-friendly form [6]. This is the advent of fish-finding devices and recreational-grade sonar systems. These systems are affordable for the public as their price, complexity and size is several orders of magnitude smaller compared to conventional sonars while the reduction in measurement accuracy has not been so significant. With the help of these devices, the classical fields of applications of the sonar systems can be left behind, and they can also be used in shallow and/or fast-moving rivers, and in small lakes. During their development sonars have experienced the classical stages of electronic improvement: (1) military application, (2) professional application, (3) affordable, widely available, sometimes recreational applications. These simple recreational-grade sonars are recognized by multiple fields of hydrologists, and several papers have been published on their applicability, accuracy assessment and how to process sonar outputs [7–15]. The topics are covered by these publications are heterogeneous, but one aspect is common: every publication deals with the reflectance or the imaging of the bed, and/or the bathymetry measurement is limited to the area under the sonar, with traditional single-beam approach. These methods and apparatus are not applicable for a large-scale bathymetric mapping mission. To survey the bathymetry of larger areas (even in rivers or in lakes) the multi-beam and the interferometric sonars are the ideal tools [16]. Thanks to the ongoing development of interferometric surveying technology, this approach becomes available also in recreational-grade fish-finders. Eventually, some fish-finders are able to take bathymetric measurements in the aperture of 140–170° on the both sides of the sonar and to create a dense cross-section-like survey [17].

During the current study, our motivation was to answer the following questions:

1. Are these interferometric, recreational-grade sonars able to create a hydomorphologically correct, underwater elevation model of shallow lakes/rivers?
2. Are these systems able to survey relatively large areas, like longer sections of a river?
3. Does the application of interferometric recreational-grade fish-finder sonars have any advance over the classical cross-section or single-beam sonars?
4. Is it feasible to build an efficient survey system based on interferometric fish-finders?

If the answers are "yes", or at least for some of the questions, then we get a novel surveying technique which is affordable, less vulnerable and applicable even at low water levels. Thanks to these properties, the measurement is repeatable, and the change of the riverbed and the lateral movement of the riverbanks is detected. A fast river bed survey method is also crucial to support other types of research, like sediment transport measurements, navigational mapping, creating the cadaster of dangerous objects in shipping routes, supplementary support of water management and hydrological engineering and riverine tourism activities. Our motivation—secondarily—was also rooted in the shortcomings of traditional river survey methods in the case of 2D hydrologic modeling. The classical, geodetic cross-sections are perfect for a small section of a river, and this was satisfactory for hydraulic engineering and 1D modeling purposes. However, with the advent of 2D modeling, it is possible to

use high-density point clouds as bottom elevation for obtaining bathymetry of higher resolution. In this case, the classical geodetic cross-sections cannot provide sufficient point density and bathymetric accuracy due to interpolation errors.

### 1.1. Evolution of Sonar Systems

The reflecting, longitudinal, mechanical waves are widely used for remote sensing purposes since the first third of the 20th century for bathymetry, primarily on seas [18,19]. Based on sound emission, sonar systems have two major types: active and passive sonars. Hereinafter, in this paper we are going to deal only with active sonars, applied for depth measurements in water bodies.

The currently used active sonars can be classified into three major groups [16]:

1. Single-beam sonars
2. Multi-beam sonars
3. Sidescan sonars, with to sub-groups:

    a. Traditional, symmetrical sidescan sonar for imaging purposes.
    b. Interferometric sonars, based on the measurement of phase difference; capable for wide-swath, 3D bathymetric measurements with wide aperture.

Our research presented in this paper is based on type 3.b., which is able to outperform multi-beam sonars in shallow aquatic environments.

### 1.2. Interferometric Bathymetry Calculations in Recreational-Grade Sonar Systems

Interferometric sonars can be approached theoretically from the niche of sidescan sonars. These sonars can be converted easily into a bathymetric survey device if we add two or more hydrophones with phase registration on an inclined baseline (Figure 1).

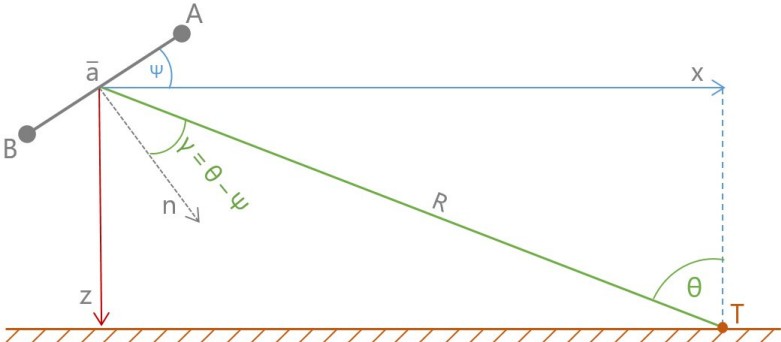

**Figure 1.** After [20], edited. '*A*' and '*B*' are the hydrophones; '$\bar{a} = \overline{AB}$' is the length of the two-element hydrophone array; the '$\Psi$' is the inclination of the hydrophone array measured to the horizon (usually 30–60°); '*T*' is the target; '*R*' is the distance of the target (hypotenuse, based on traditional sonar measurement); '*n*' is the normal vector of the '$\bar{a}$' section; the '$\theta$' can be expressed geometrically based on the following equation: '$\theta = \gamma + \Psi$'; '*x*' and '*z*' vectors are the orthogonal components of '*R*'. The '+' direction points right and downwards.

Interferometric sonars can be approached theoretically from the niche of sidescan sonars. These sonars can be converted easily into a bathymetric survey device if two or more hydrophones with phase registration functionality on inclined baselines are added (Figure 1).

If we know the length of the $\bar{a} = \overline{AB}$ section, based on the actual design of the interferometric sonar (Figure 1), the particular length of the '$R$', the inclination of the sonar array '$\Psi$', we can calculate the '$\Delta\varphi_{AB}$' phase difference with the following equation [20]:

$$\Delta\varphi_{AB} = k\delta R = ka\sin\gamma = 2\pi\frac{\bar{a}}{\lambda}\sin\gamma, \tag{1}$$

where '$k$' is the "key number" $\left(k = \frac{2\pi}{\lambda}\right)$; '$\lambda$' is the wavelength in the particular medium; '$\gamma$' is the angle in Figure 1 ('$\gamma = \theta - \Psi$'); '$\delta R$' is the length difference between $\overline{AT}$ and $\overline{BT}$ sections (Figure 1).

From this Equation (1), it is possible to express the '$\theta$' angle on demand [20]:

$$\theta = \sin^{-1}\left(\frac{\Delta\varphi_{AB} + 2\pi\cdot n}{ka}\right) + \Psi, \tag{2}$$

where $n \in \mathbb{Z}$ ($n = \cdots -2; -1; 0; +1; +2\cdots$; see below). With the help of the '$\theta$' angle, if we know the length of '$R$' (from traditional sonar measurement) we can get the length of '$x$' and '$z$' vectors in Figure 1.

Unfortunately, '$\Delta\varphi_{AB}$' cannot be measured directly as a real phase difference: the two hydrophones are able to provide the *apparent*, *relative* phase difference [16,20].

In a particular case, if the phase on the hydrophone '$A$' is $\frac{1}{4}\pi$ and the phase on hydrophone '$B$' is $\frac{3}{4}\pi$ the phase difference is inevitably $\frac{2}{4}\pi$. However, due to various geometric constellations, it is also possible that the real phase difference is more than $2\pi$. See another practical example in Figure 2:

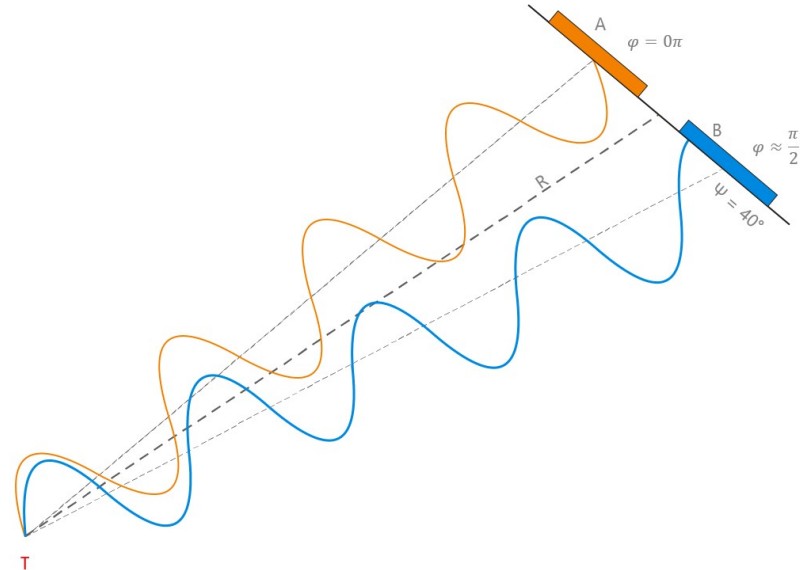

**Figure 2.** Practical detection of phase difference on a hydrophone array. See Figure 1 for the letters. The phase difference can be calculated with the following equation: $\Delta\varphi_{AB} = \varphi_B - \varphi_A = \frac{\pi}{2} - 0 = \frac{\pi}{2}$. Due to the reflection, the phase in point '$T$' is always the same. The wavelength and the size of hydrophones are not in scale. The sinewave symbolizes the displacement of the particles in the longitudinal wave burst. After the idea described in [21].

In certain implementations, hydrophones are only able to detect the apparent phase in the returning pulse. To counteract this fact, we should introduce an '$n$' variable to symbolize the multiple solutions of the Equation (2). This introduces a serious uncertainty in our calculations: for the same '$\Delta\varphi_{AB}$' value we could get several '$\theta$' angles, and the erroneous solutions should be filtered out by the internal electronics of the sonar system in consideration of the neighboring points in the given swath. After the application of this post-processing algorithm, the recreational-grade sonar provides us the '$x$' and '$z$' vectors (Figure 1).

Unfortunately, even if we choose the correct value of '*n*' in Equation (2), there are still some inaccuracies in the calculation. Around the normal vector of '$\bar{a}$' section on Figure 1, the inaccuracy is low, but as the measurement moves either to the left or to the right side of the normal vector, the error increases gradually, forming a butterfly shape of uncertainty [22,23] The amount of error can be expressed by the following equation [20]:

$$\frac{\delta z}{z} = \frac{\delta \Delta \varphi_{AB}}{2\pi} \cdot \frac{\lambda}{\bar{a}} \cdot \frac{\tan \theta}{\cos \gamma} \tag{3}$$

See Figure 1 and Equation (1) for letters. It is useful if the final raster is interpolated by an algorithm, like Empirical Bayesian kriging, which accounts for the errors in the calculation procedure [24]. If we know the predicted amount of measurement error, we are able to filter out all marginal points with unreliable readings.

## 2. Materials and Methods

Due to the limitations of traditional riverbed surveying methods, a unique solution for bathymetric measurements had to be developed for recreational-grade interferometric sonars. From the construction of the carrier platform to the survey plan, our development approach was divided into six major steps:

1. Selecting the carrier platform, to minimize acoustic disturbances caused by the cavity and the boat engine; reduce the pitch, roll and heave movement of the hydrophones of the sonar.
2. Selecting, then building an adequate sonar system, based on a recreational-grade sonar and its auxiliary devices and on a geodetic Global Navigation Satellite System (GNSS).
3. Create a suspension to use the sonar securely. The suspension should resist collisions caused by the undetected floating objects and logs. In cases of emergency, the transducer should be removed from the water immediately.
4. Processing of recreational-grade sonar output formats has never been common. To gain maximum control we developed our own sonar processing application.
5. Create an effective measurement plan to maximize sounding quality.
6. Bathymetry data alone is insufficient to characterize channel morphology: geodesic survey of the neighboring areas should be included, as well as the mapping artificial hydroengineering structures.

### 2.1. Carrier Platform

Our carrier platform was a double-hulled, geometry-stabilized, wide-beam, low-draft, pure aluminum alloy catamaran boat with the following dimensions: $6.0 \times 1.2 \times 2.3$ m (L × H × W). At the bow, the entrance-edges of the hulls were rounded backwards, towards the water line. A protruding crossbeam framework connected the two hulls of the boat. This structure allowed the installation of the sonar system on a relatively stable, meanwhile low draft carrier [25]. With this design, our primary goal was to minimize the roll of the vessel. Due to the low draft and the backwards rounded hull we were able to approach the riverbanks securely. The propulsion of the boat was provided by a 50 HP Yamaha™ outboard engine (Yamaha Corporation, Hamamatsu, Shizuoka, Japan), mounted on the center of the transom beam.

### 2.2. Sonar System and Auxiliary Devices

As a first step, the sonar system was selected. We choose the Lowrance® (Tulsa, OK, USA) sonar brand (subsidiary of Navico Inc., Tulsa, OK, USA) because at the beginning of our research in late 2016, the Lowrance® was the only provider of affordable recreational-grade interferometric fish-finder sonars, but meanwhile other companies like Furuno® (Nishinomiya, Hyōgo, Japan), Garmin® (Schaffhausen, Switzerland), Humminbird® (Johnson Outdoors Inc., Racine, WI, USA) and Raymarine® (Portsmouth, United Kingdom) made a huge step forward.

The schematic design of the selected sonar system is presented in Figure 3:

**Figure 3.** The schematic structure of the applied Lowrance® sonar system and auxiliary devices. The fish is just for illustration purposes. See detailed description in the text below.

The central part of our sonar system was a Lowrance® HDS–7" Gen3 Touch touchscreen unit (OS Ver.: 5.0 – 57.1.219; 2nd row, 2nd column, in the center, on Figure 3). This device is functionally a rugged tablet with a factory modified Linux distribution. This unit serves as a display, a data recorder and as a primary user interface. The HDS module was equipped with a National Marine Electronics Association (NMEA) 0183 connector.

Via this connector, a one-way communication link was established with the geodetic Real-time Kinematic GNSS (RTK+ GNSS; GeoMax® Zenith35™ Pro, made by GeoMax AG, Windau, Switzerland, 10 Hz; 1st row, 1st column). This GNSS provided the location readings, as the inbuilt (and less precise) GNSS of the HDS unit was switched off.

The HDS unit also includes a NMEA 2000 connector. This interface had multiple purposes:

- The primary function was to establish a connection with the solid-state tri-axial gyroscope and magnetic compass (Lowrance® Precision–9, SW Ver.: 9.2; 1st row, 2nd column). With the readings of this gyroscope, the HDS module was able to counteract the vertical displacement of the interferometric bathymetry readings—even in case of extreme roll of the platform.
- This connector serves as the connection point of a submerged paddlewheel water speed and water temperature sensor. (Lowrance® EP–70R; 3rd row, on the right side).
- The tertiary function of NMEA 2000 is to connect a broadband radar device (not used in this research).

The major data collector is an integrated Lowrance® sidescan, interferometric and DownScan™ transducer (Lowrance® StructureScan® 3D; 3rd row, on the right side; see detailed description of this transducer in [21,26,27]) which was connected to the central HDS unit through a signal processor (2nd row, 3rd column, with "SS3" script on it). An additional single-beam transducer (Lowrance® HST–WSU 83/200 kHz Skimmer Transducer) was also used to get primary depth information (not presented separately in Figure 3).

A standard 12 V Ca–Ca car battery (2nd row, 1st column) supported the power supply of the system.

All single-beam, interferometric, sidescan and DownScan™ readings are recorded by the HDS unit in SL3 format [28].

*2.3. Setup of the Measurement System*

At the bow, on the protruding crossbeam of the catamaran, a 3-meter-long steel rod was installed at the centerline of the boat between the two hulls. The bottom of the rod was submerged under the surface of the water. At this end there were the sensors installed with the draft of 0.4 m (Figure 4).

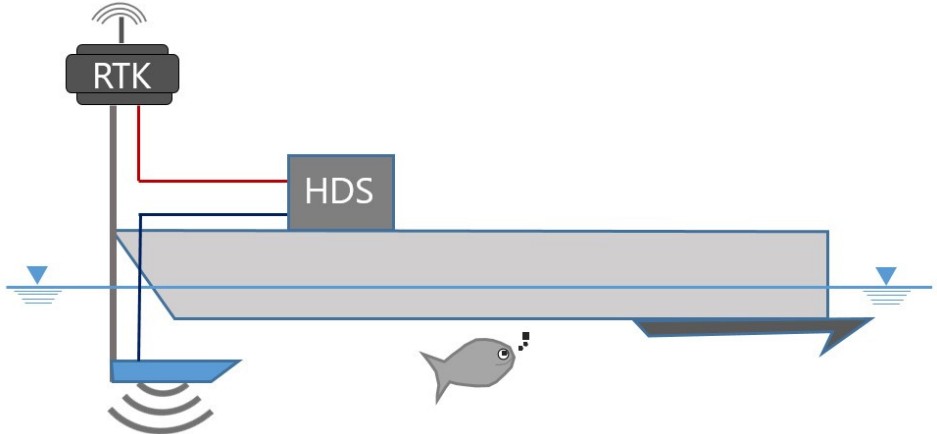

**Figure 4.** The schematic diagram measurement setup. The sizes and the devices are not to scale. See Figure 3 and the text above for the captions.

The sensor array includes an integrated interferometric-, sidescan- and DownScan™ transducer, a single-beam transducer and a paddle-wheel water speed sensor. With this implementation, the transducers were the first objects that cut the water (Figure 5).

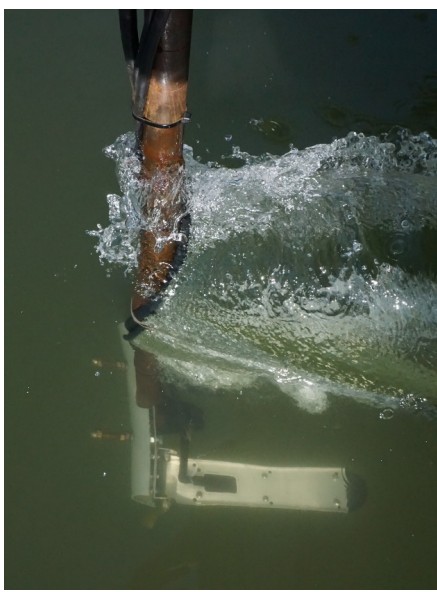

**Figure 5.** The sensor array during a measurement campaign. The sensors are the first objects that cut the water.

This placement helps to reduce the hull- and engine-induced noises, turbulence and cavity [8,29,30]. With this implementation, we were able to maximize the accuracy of the interferometric measurement and get the most detailed sidescan and DownScan™ images without interference and noise strips.

At the other end of the rod, pointing to the sky, a geodetic GNSS was mounted on a standard 5/8″ bolt.

This placement of the sonar effectively reduces the cavity-induced noises, however, at the same time the possibility of a collision with floating or submerged objects or dead logs was highly increased. To solve this problem, a rod with sonar in its bottom was mounted on the bow which can be tilted backwards onto the bow in the hazard of collision with any floating object. However, not all collisions can be avoided this way: therefore, a zinc-coated steel plate covered with a PVC semi-cylinder with two fenders were also mounted on the rod to protect the sensors (Figures 5 and 6).

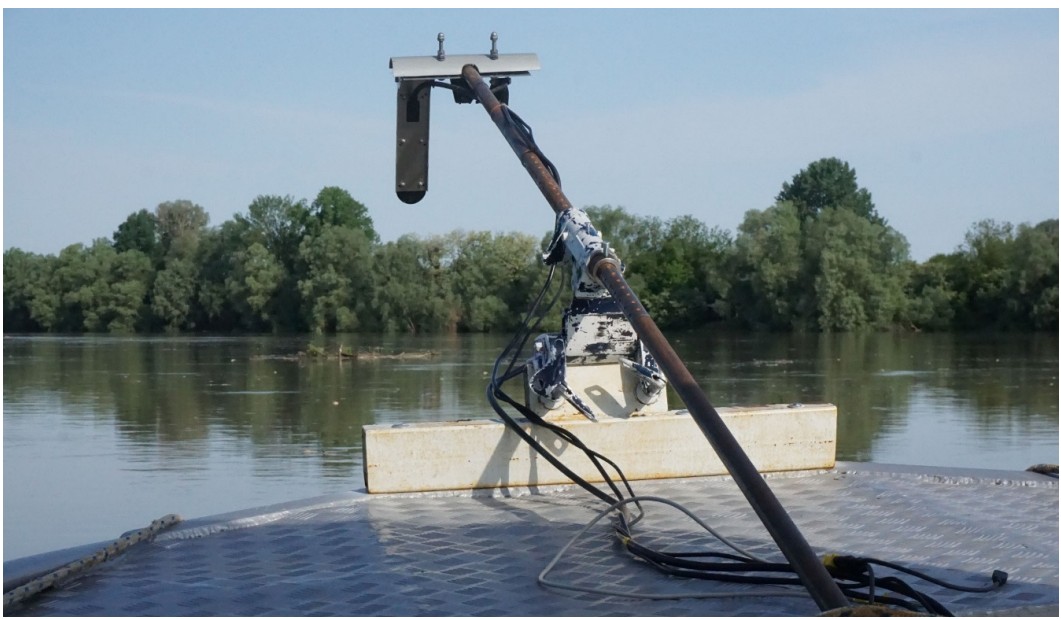

**Figure 6.** The placement of the sensor array at the bow. The rod can be tilted back to avoid collisions with floating logs. A protective cover is mounted in front of the sensor array to minimize the damage caused by small floating objects.

*2.4. Software Environment for Data Processing*

The HDS unit can store the sonar readings in three file formats: SL3, SL2 and SLG. These sonar log files are proprietary data formats, which were created and maintained by Lowrance®. Due to these circumstances, we were unable to find a ready-to-use, complete open-source software to process these files. Therefore, we were to develop our own a SL3 file processor. This SL3 processing software was written in C♯® and partially based on own experience in the field of interferometric processing and in other aspects based on the knowledge available in these articles and software: [11,12,16,17,31–37].

The developed software is able to:

1.  Open the SL3 files and restore the trajectory of the ship or boat; and overcome the limitation of Lowrance's Mercator-like projection and restore the accuracy based on GNSS readings with the auxiliary "USR" files [32]. This trajectory includes millisecond level timestamp 'X' and 'Y' coordinates in a projected coordinate system, actual speed readings and actual heading (based on the trajectory of the GNSS and based on the solid state triaxial gyroscope (Lowrance®, Precision–9) to detect drift-like movements). The output point set is stored in File Geodatabase format, made by ESRI (Environmental System Research Institute, Redlands, CA, USA).
2.  Export sidescan imagery into perspective and georeferenced 8-bit Tagged Image File Format (TIFF) files, and export these readings into point cloud (File Geodatabase)
3.  Export single-beam and DownScan™ imagery into 8-bit TIFF files.
4.  Export 3D bathymetry readings into point cloud (File Geodatabase). This dataset is based on the processed interferometric measurements of the StructureScan 3D transducer.
5.  Filter erroneous readings, originated either from the GNSS or from the interferometric measurements.

### 2.5. Processing of the SL3 Files

Every measurement made by a Lowrance® HDS unit can be saved into SL3 files. The original purpose of this functionality was to provide a playback option for the anglers to analyze the morphology of the lake or river bed and fish behavior. However, with careful data processing, acquisition of high-quality and science-grade data from the sonar logs is enabled [12].

The processing method of these files is not standardized: the format of the file is closed, proprietary and binary and the file format is managed and authored by the owner of Lowrance® (Navico Inc., Tulsa, OK, USA). There are some closed source, commercial, out-of-the-shelf applications which were able to process these files (ReefMaster®, developed by ReefMaster Software Ltd. Birdham, UK; SonarTRX®, made by Leraand Engineering Inc., Honolulu, HI, USA), but the capabilities are usually focused (and limited) to the needs of the recreational fishing society. The main output of these applications is the cleaned or normalized sidescan images which are not focused on scientific data outputs. Until today, we are unaware of any commercial and recreational-grade applications that capable to convert interferometric bathymetric data of recreational-grade sonars to GIS formats.

Today, SonarWiz® (Chesapeake Technology, Inc., Los Altos, CA, USA) is the only, highly professional software, which is able to read almost all data type from an SL3 file. However, if we apply this application we lose the control over the export procedure: data output is somehow cleaned, post-processed and altered, and these services cannot be switched off, meanwhile the source code of these features is closed. The software is also rather expensive and, in this case, we lose the freedom provided by the application of the recreational-grade devices.

We also checked the opportunities of open-source programs: the software PyHum made for Humminbird® devices was found as the only software specifically developed for scientific purposes [12]. However, this application is used for sidescan analysis only, i.e., was designed for an another recreational-grade sonar system. To overcome the aforementioned challenges, we developed a software specific to our measurement purposes.

After the examination of the data structure, we identified the following datasets:

1. Conventional, downwards looking single-beam sonar datasets.
2. Elliptic, high frequency, downwards looking single-beam sonar dataset.
3. Sidescan datasets, to understand the morphology and textural pattern of the channel.
4. Interferometric, 3-dimensional bathymetric packages for elevation data.
5. We also identified two additional datasets. They are related to the raw data of the hydrophones and participating in the creation of 3D model, probably in such way, which is mentioned in [3].

SL3 files are containers for multiple sonar readings. The original intention of this file format was to provide a simple file format to record the location and the properties of fish habitats to guide anglers back to the prosperous fields. The recorded products range from single-beam readings, sidescan readings, interferometric measurements to DownScan™ readings. The SL3 files can be played on the HDS device like a movie—based on different sonar products. To provide acceptable playback experience on low-performance devices the sonar readings are organized into datasets called 'frames'. These frames contain one column of the actual sonar product, like one column of sidescan, followed by one column of DownScan™, followed by a sidescan frame again.

Every frame has a header with the following properties:

- Millisecond-based timestamp.
- Speed over ground (measured in knots, based on the 10 Hz GNSS readings sent via NMEA 0183, floating point).
- Water speed (based on the readings of the paddlewheel sensor connected via NMEA 2000, floating point).
- The true heading (true course over ground) of the movement (based on GNSS readings, floating point).

- The magnetic heading (based on the magnetically referenced Lowrance® Precision–9).
- The elevation of the GNSS device, measured in feet over WGS84 (World Geodetic System 1984) ellipsoid (floating point).
- Position of the GNSS, recorded in integer numbers, projected to a metric grid (see below).

However, the connected geodetic GNSS provides 'λ', 'φ' information measured with high precision on WGS84 ellipsoid via the NMEA 0183 protocol. The coordinates were transformed by the following equations into a simplified Mercator-like projection, called Lowrance Mercator (SR-ORG: 8230)—managed by Lowrance®:

$$\left\{ \begin{array}{c} X = R \cdot \frac{\lambda \pi}{180°} \\ Y = R \cdot \ln\left[\tan\left(\frac{1}{2}\left(\frac{\varphi \pi}{180°} + \frac{\pi}{2}\right)\right)\right] \end{array} \right\}, \tag{4}$$

where the '*R*' is the polar radius of the WGS84 ellipsoid (which was truncated to 6 356 752.3142 m [32,33,35]); the 'λ' is the geographic longitude (over WGS84); the 'φ' is the geographic latitude (over WGS84). Although the measurement of the coordinates was accurate during the field surveys, due to the recreational-grade level of the electronics and file format, each coordinate value was truncated into integer numbers which could generate a maximum error of $\sqrt[2]{2}$ m (1 × 1 m Manhattan grid) plus the error coming from the GNSS device itself. To overcome the error, we had to use the unbiased, floating-point values from the header to restore the original precision of the coordinates.

With the blending of these data, we were able to restore the position of the transducer approximately, with a better precision than $\sqrt[2]{2}$ m. The approximation was calculated by the following equations:

$$\left\{ \begin{array}{c} \alpha = \frac{1}{2} \cdot (\alpha_0 + \alpha_1) \\ v = \frac{1}{2} \cdot (v_0 + v_1) \\ t = t_1 - t_0 \\ s = v \cdot t \end{array} \right\}, \tag{5}$$

$$\left\{ \begin{array}{c} X_1 = X_0 + s \cdot \cos \alpha \\ Y_1 = Y_0 + s \cdot \sin \alpha \\ Z_1 = Z_{1 \text{ GNSS}} \end{array} \right\}, \tag{6}$$

where '$\alpha_1$' is the heading stored in the actual data frame, '$\alpha_0$' is the heading stored in the previous data frame; '$v_0$' is the previous speed (over ground), the '$v_1$' is the actual speed (over ground); '$X_0$' and '$Y_0$' are the initial coordinates.

This calculation provides a better approximation, but due to the nature of the equations a possible error could spread through the whole trajectory. This spreading was limited by external 'λ', 'φ' readings, recorded by the HDS unit in a separate track file called USR, alongside the SL3. This file contains unprojected, high-precision, timestamped waypoints in every second.

The augmented trajectory and the USR readings were compared in every second and the observed error was distributed by time weighting in every one-second period. This helps to blend a high frequency, but originally low precision trajectory with a low frequency, but high precision trajectory to achieve maximal positional accuracy during the measurement campaign.

Figure 7 shows the original readings (in Lowrance Mercator) provided by the sonar itself. The approximately restored trajectory of the boat, based on the considerations mentioned above, is shown in Figure 8.

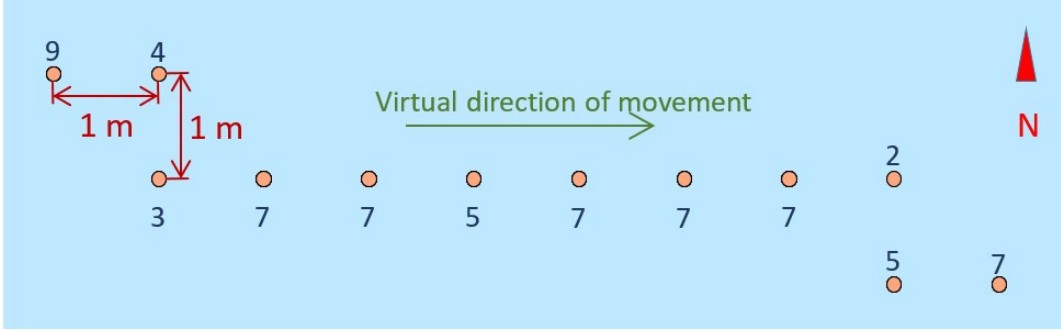

**Figure 7.** Dilution of precision caused by the truncation of projected coordinates in Lowrance devices.

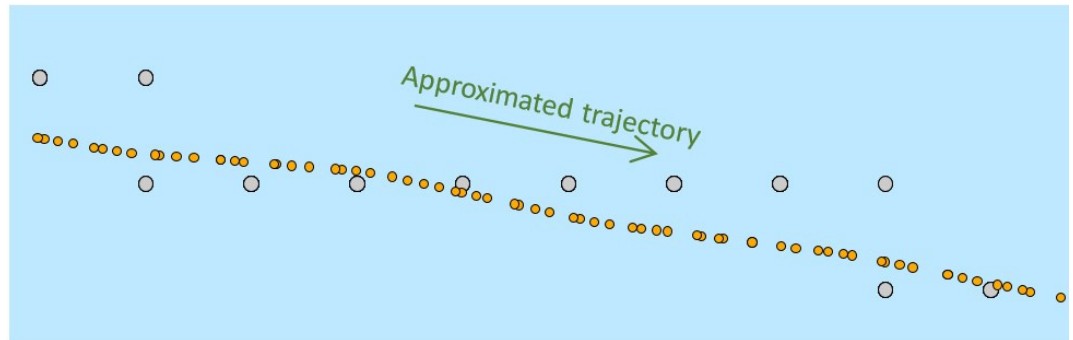

**Figure 8.** The improved trajectory of the vessel using the augmented precision algorithm.

Nonetheless, these augmented precision trajectories still required some post-processing: the applied GeoMax® Zenith35™ Pro is a high-precision survey system, but during the survey, smaller glitches and disturbances could occur. These glitches could be caused by human error, physical loss of GSM/UMTS (Global System for Mobile Communications/Universal Mobile Telecommunications System) signal and the temporary malfunctions of the RTK+ system.

During the data processing, we were unable to identify these error sources (neither Dilution of Precision, nor measurement method readings were recorded in the files) and due to the lack of continuous reference GNSS measurements we could not specify the absolute value of error.

To overcome this problem, we introduced a few automated quality checks to detect unforeseen measurement errors. As a first step, we filtered out the gross errors. The 'height' component of the GNSS measurement is largely affected by measurement errors, so we chose this property to detect serious discrepancies. In consideration of the geographic environment, if the 'height' value was less than 100 meter above WGS84 ellipsoid or more than 280 meters above the same reference system the measurement point was rejected automatically. In this filter, we also filtered out readings where the either the 'λ', or the 'φ' readings were zeroed out by the GNSS itself.

As a secondary filtering procedure, we checked the relative height during the measurement campaigns. As no measurement campaign was longer than 10 km and the river surface is approximately leveled, we marked the measurements where the height of the actual reading was shifted by 0.5 meter below or above the median height of the actual measurement campaign.

In our software environment, there were two options for the management of these marked readings:

1. Remove them completely, but in this case uncovered areas appear in the bathymetry.
2. Correction with linear interpolation (or extrapolation at the extremities) based on their neighbors and assuming correct spatial data, hence the river surface is almost leveled, with minute downstream slope. Due to this procedure, several measurements are avoided, but likely, an error of 0.01–0.15 meters is introduced. In this case, the reduced precision in horizontal direction remains undetected and could cause a small 'zigzag' pattern in the final trajectory. The horizontal

measurement errors usually appear jointly with the 'height' errors. However, after several trials, we found that these errors had a smaller effect on the final bathymetric product than the rejection of these points and exclusively relying on an interpolation algorithm.

Not only the position of the points need some post-processing: the course (course over ground; calculated by the GNSS) and the heading values (calculated by the Precision–9 gyroscope) need some filtering too. These values have a primary importance as the interferometric measurements are projected exactly to left and to right, perpendicular to the trajectory. As a result of the high sampling frequency of the sonar and the lack of the built-in course over ground variation reduction function, even a very tiny GNSS measurement error, or the rolling movement of the vessel may cause a detectable erroneous fluctuation in the course. This phenomenon has a serious consequence: while the boat has a straight trajectory and the measurement error of the GNSS was considerably low, the bathymetric swaths still have a false " 𝕎 " (crisscross) pattern. To eliminate this error, we introduced a vector based, 3-point-long moving average method. In the algorithm, we took the previously corrected coordinate of points '$n^{th}$' and '$n+2^{th}$' and form a vector pointing from '$n^{th}$' to '$n+2^{th}$'. We assigned this vector to point '$n+1^{th}$'. This sampling is long enough to clear the random fluctuations, but small enough to preserve real bends in the course. With this modification the theoretical/ideal "|||||" look of bathymetric measurement can be approximated.

After dampening the magnitude of errors we were able to calculate the specific bathymetry of the channel. In this particular implementation of Lowrance® StructureScan 3D sonar, in every interferometric data frame we obtained an array of '$x$' and '$z$' vectors (Figure 1). The readings were grouped according to their portside or starboard positions. The first reading was on the left side of the sonar at a distance of 0' (the length of '$x$' is 0 m), followed by the '$z$' vector with the actual depth immediately under the transducer. The second '$x$' reading was on the left, 0.3048 m (1') away from the sonar, followed by the next '$z$' value. This pattern continues regularly with a horizontal distance increments of 0.3048 m until the reading is considered unreliable by the firmware of the sonar. If the reading is unreliable or missing by geometrical reasons either the next reading could jump several feet away, or the sonar starts a new array at the center moving to the right side. The number of readings are not necessarily equal, and the distribution could be asymmetric [17]. This data structure is interpreted as a cross-section, and the interferometric bathymetry survey itself can be understood as a dense series of cross-sections with a large number of points. The individual points in each cross-section can be calculated as follows:

$$\left\{ \begin{array}{l} X_b = X_c \pm x \sin \alpha_c \\ Y_b = Y_c \pm x \cos \alpha_c \end{array} \right\}, \tag{7}$$

where '$X_b$' and '$Y_b$' represent the coordinates of the actual point; '$X_c$' and '$Y_c$' represent the corrected coordinates of the sonar; and '$\alpha_c$' is the corrected heading. The height of the river bottom can be calculated directly with the help of '$z$' vector (Figure 1) based on the actual GNSS height.

## 2.6. Measurement Plan

The survey area was approximately 128.6 km long (72+000 → 198+600 rkm) section of the River Drava. The area of interest, with the adjoining oxbows, covered an area of ~36 km². The maximum width of the main river reached ~230 m, the minimum width was ~105 m. In terms of the swath (2 × 70°) we took at least 5 runs to obtain a comprehensive bathymetric coverage in the main channel, while the average depth was 2–4 m. We divided this area into 35 sections of 4–5 km length (with some overlaps).

Based on practical considerations, the stability of the boat and the amount of cavity-induced disturbances visible on sidescan images measurements were only carried out in downstream directions at a speed of 7 to 10 km/h (speed over ground). See a detailed description of the plan in [38].

*2.7. Data Integration*

In addition to channel bathymetry, bank topography data is also essential when a digital elevation model, suitable for hydrologic modeling, is generated. For the topographic survey of the immediate 25-meter-wide area along the river classified Light Detection and Ranging (LiDAR) data were obtained. The transition between the riverbed and the bank is rarely smooth: we digitized semi-automatically the orthophotography-based river banks at bank-full discharge. Due to the specific hydromorphological properties of the Drava River the bank contours of 147 islands were also digitized.

For the final interpolation, the following datasets were used:

1.  Single-part point data derived from the sonar readings. This is the major data source for riverbed interpolation. This dataset was stored in File Geodatabase. The Lowrance® sonars use a simplified horizontal reference system, called "Lowrance Mercator" which is based on a custom sphere. This reference system was transformed into EPSG 23700. The vertical reference system of the original sonar readings was based on the heights above the WGS84 geoid. These heights were then transformed into EPSG 5787 via the Lechner Knowledge Center's transformation protocol.
2.  Contour lines of bank-full discharge. This dataset included the primary break lines for riverbed delineation. The 3D lines were transformed into high-density point datasets to avoid the "leaking" of the interpolation algorithm.
3.  Lines delineating the islands. Every island was delineated using the same steps mentioned above.
4.  LiDAR dataset of the research area. Every elevation reading comes from this LiDAR inside the islands and outside the banklines.

Geodetic heights of artificial objects in and around the river (embankments, bridges, etc.).

After several experiments the ANUDEM (Australian National University Digital Elevation Model; "Topo To Raster" geoprocessing function in ArcGIS® Desktop, [39]) was used to interpolate these datasets with no drainage enforcement. This way a smooth continuous digital elevation model was obtained.

## 3. Results

After the iterative process of development, the Hungarian reach of the Drava River was chosen for experimental purposes. The Drava is a right-bank tributary of the Danube [40]. The Drava River has a total length of 725 km from its source in Northeastern Italy in the Dolomites and to its confluence with the Danube at Aljmaš in Croatia [38].

Our research area covered a significant portion of the Hungarian reach of the Drava from 75+000 to 198+600 rkm (Drávaszabolcs, Hungary to Vízvár, Hungary). The studied reach was long enough to test our system and evaluate time and labor requirements for a rivers of similar size. During the survey, 222 separate SL3 files were recorded, in the total size of 85.8 GiB (uncompressed). For an easier processing, recordings were separated into ~5-km-long sections. For these individual sections of an ESRI File Geodatabase was produced with the following outputs:

- Trajectory of the boat in original, Lowrance Mercator projection, with 1-meter resolution and with restored precision (in separate feature classes). The route points were classified into primary (single-beam), DownScan™ (single-beam with narrow elliptic footprint, perpendicular to the trajectory), sidescan and interferometric 3D measurements. All route points possessed information on (1) measurement time (in UTC and in milliseconds since the beginning of the recording of the actual SL3 file); (2) flow velocity; (3) speed over ground (GNSS based); (4) the GNSS based heading; (5) magnetic heading (corrected by the declination); (6) water temperature; (7) the water depth under the transducer (measured by the single-beam sonar); (8) the minimum and maximum range of the sonar for the particular measurement type, based on the actual water depth; and (ix) a flag about the validity of the GNSS-supplied position. The 'X' and 'Y' coordinates were recorded in Lowrance Mercator; the 'Z' coordinate was recorded as the height over the WGS84 geoid. Every point was stamped with a record number for cross-processing with other datasets.

- Interferometric bathymetry measurements, in a multipoint feature class. This dataset contains cross-section-like bathymetric points perpendicular to the trajectory. Due to the limited accuracy of the sonar over a swath angle of 70° every point was rejected over this value automatically; so the effective aperture is 140° (theoretically, if the bottom is flat). Every measurement group was stored in one database row, with multiple points, and every point group is stamped with the record number (see above). The projection of the dataset is Lowrance Mercator. When our software calculates the coordinates of the points based on the '*x*' and '*z*' vectors (Figure 1), provided by the sonar, it takes the augmented trajectory line as primary coordinate. The 'Z' values of the points are bases on the GNSS height measured over the WGS84 geoid, and it was corrected by the length of the suspension rod. In this case the elevation information of the multipoints of the dataset are in absolute height values. This helps us to eliminate the issues caused by changing water level, and the changes in the draft caused by the different loads on the vessel. This was the primary dataset used for further processing.

- Sidescan measurements in a multipoint feature class. Due to the digital manner of the SL3 files, sidescan readings can be represented as individual points in 2D space. The structure of this dataset is similar to the interferometric measurements: the point of origin is positioned on the augmented trajectory line, but the displacement of the individual points is based on the range reported by the sonar. These points have no 'Z' information, hence they are positioned in the bottom of the channel, but the reflectance value is encoded as an 'M' property. This reflectance value is measured bytes, ranging from 0 to 255. Every cross-section-like point group has 3200 points based on the design of the SL3 file. These measurements are significantly distorted at their margins, closed to the river banks. The horizontal reference system is Lowrance Mercator.

- Geometrically corrected sidescan measurement in multipoint feature classes. This dataset has the same properties as the previously described one, but the points are corrected to the plane bottom, by simple Pythagorean theorem. The actual implementation for the sidescan sonar systems is described in [16]. In other software, able to export georeferenced, geometrically corrected sidescan images, the output is raster based. However, the distribution of the measured points does not correspond with a raster geometrically. Our primary idea was to use these points as an input of interpolation, to eliminate the distortions caused by the idea of the raster itself. Additional processing possibilities for this dataset are described in [41]. Similarly to [12,16], water column was removed in this dataset.

For visualization purposes, our software is also capable to export single-beam, DownScan™ and sidescan images as raster. This export function is implemented in two ways:

- In the first case, the images were exported in TIFF format, but no direct georeferencing information was included. The output images are horizontal and straight, in a consecutive order. During the survey, based on water depth, the range of the sonar is changing according to the inbuilt range categories of the Lowrance® unit. When sonar range altered a new image was started. In certain cases, the sonar omits some readings (it is likely caused by logs or other kind of floating objects or by serious turbulence). If the gap exceeded one meter, a new image was started to avoid the spread of the error over the whole image. Finally, a false georeferencing information was added to the images for visualization purposes: the images in this case were displayed in a straight manner, however the length along the trajectory line was accurate and the gaps are also displayed precisely. This false georeferencing helped to align different sonar products to each other than analyze them synoptically (usually sidescan and DownScan™). Compared to the length of an image the objects on the bottom usually look flat. To make the interpretation easier we added a transverse exaggeration of the factor of 10 through the false georeferencing. Every TIFF file was indexed on 8 bit, with a custom palette. As a last step, a Persistent Auxiliary Metadata dataset (PAM; *.tif.aux.xml) was added to the images to provide constant look through fixed statistics in different GDAL-based (Geospatial Data Abstraction Library) GIS applications

(ArcGIS® Desktop and QGIS). A section of the river bottom is presented with false georeferencing in Figure 9 (DownScan™) and Figure 10 (sidescan).

- In the second case, the georeferenced images were exported in TIFF format (Figure 11). Georeferencing was implemented through the PAM dataset. In the case of sidescan three origin-destination control point pairs were added to the first column of the image: at top, left; left, at the middle; bottom left (always in pixel center). This pattern was repeated at least in the last column of the image or after every $25^{th}$ column in the image. If the number of the control points on the given image was less than nine a standard polynomial adjustment was applied, otherwise spline was used. Some transverse exaggeration was also applied when the geometric correctness was not important in the actual situation. The ideal transverse exaggeration for sidescan imagery was 4.5.

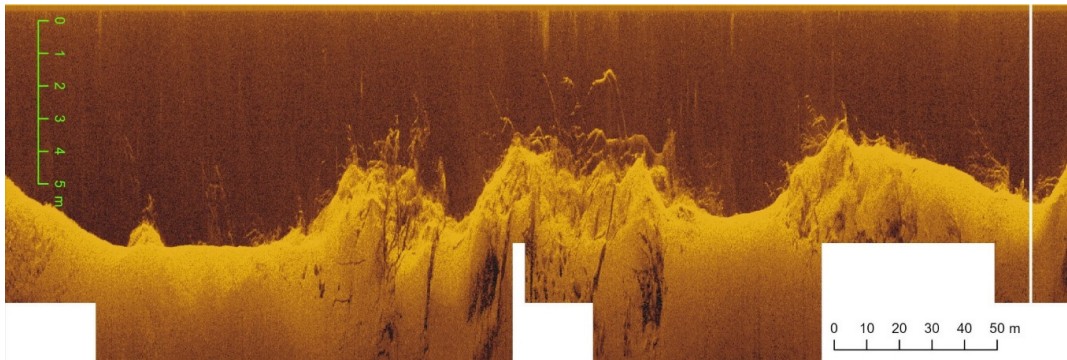

**Figure 9.** DownScan™ image at 171+000 rkm on the Drava River. The image is exaggerated vertically by the factor of 10. See the green scale bar at the left for depths.

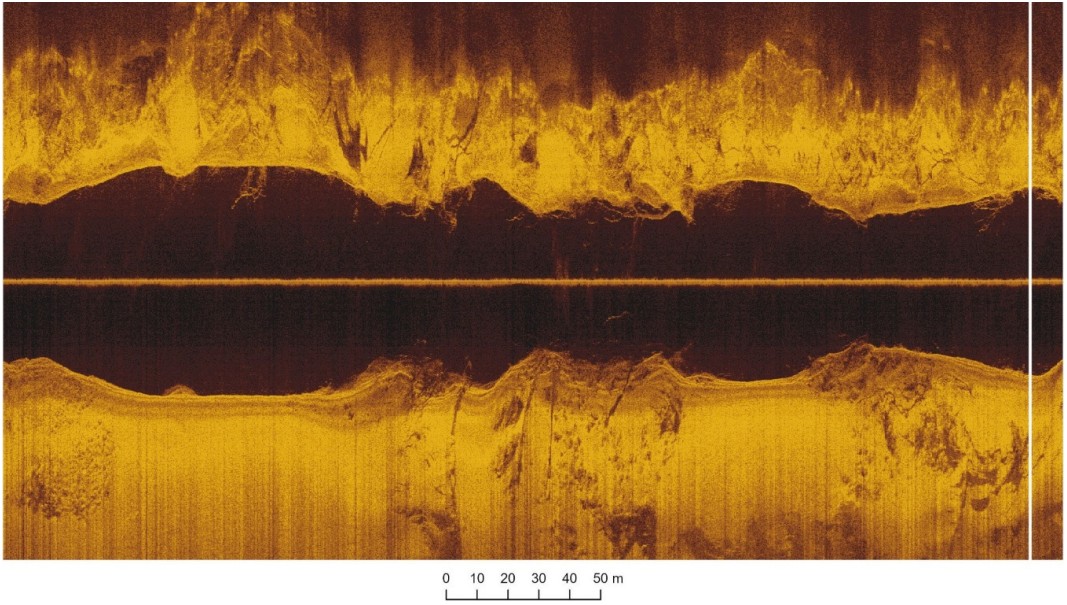

**Figure 10.** Sidescan image at 171+000 rkm. This figure and Figure 9 were generated along the same trajectory. The interpretation of the sidescan images is described in [16].

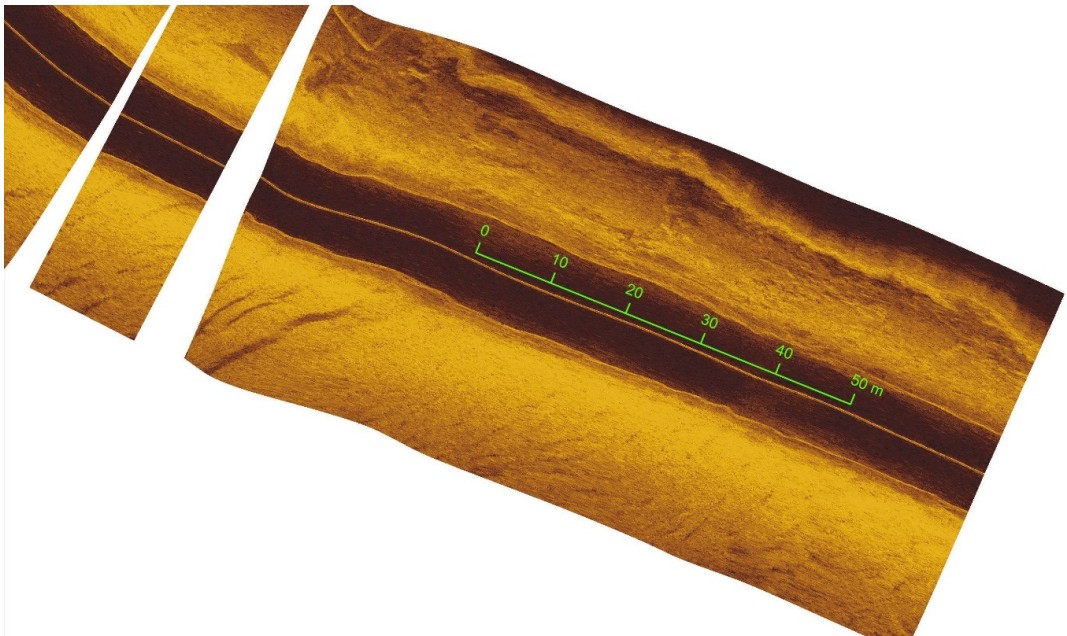

**Figure 11.** Georeferenced sidescan image at 174+000 rkm. The longitudinal length along the trajectory line is true distance, but a transversal exaggeration of 4.5 was applied to help visual interpretation. In the transversal direction, serious distortions could occur due to the topography of the bottom. The black fringe on the top of the image is caused by the river bank.

Following the data export multiple derivatives were generated. Our primary data source was the interferometric multipoint dataset. If the digital elevation model of the river bottom is generated with the immediate fringes included, a recreational-grade interferometric sonar may be used for the survey of shallow rivers. To achieve this goal, after the manual inspection of every swath, we projected every dataset from Lowrance Mercator to the Uniform National Projection System (EPSG 23700). This projection was done using ArcGIS® Pro 2.3. The WGS84 geoid heights were translated to the Unified National Vertical Network (EPSG 5787) via the official calculator provided by the Lechner Knowledge Center. We used ANUDEM to compile the sonar readings with external data sources [42,43]. We used LiDAR ground points in the 25-meter-wide area around the river in case of bank full discharge. To prevent ANUDEM from artifact generation at the extremities, manually generated elevation lines were used to describe the outlines of in-channel islands as well as the banklines of the bankfull discharge. The elevation information of engineered artificial objects was also included. All multipoint datasets were converted into a single point feature class to conform to ANUDEM. Due to the performance limitations of ANUDEM we sliced the dataset into 64 smaller, overlapping areas.

The following (other-than-default) settings were used for interpolation:

- Output cell size: 1 m; snapped to a reference DEM (Digital Elevation Model) to prevent skidding between interpolation areas.
- Margin in cells: 5. This setting is used to prevent artifacts at the beginning and at the end of the river. On intermediate areas it is irrelevant.
- Drainage enforcement: "Do not enforce". ANUDEM tries to create sink-free elevation models if no sink feature was provided. However, the underwater conditions do not correspond with land conditions, so the drainage enforcement should be avoided.
- Primary type of input data: "Spot".
- Discretization error factor: 1.4. This slightly enhanced value is used to smooth smaller fluctuation.

- Vertical standard error: 0.1 m. In most cases, the pixels are oversampled by bathymetry data. This setting helps the ANUDEM to clean outliers in the group, caused by measurement error, or accidental events, like transiting logs in the water column.
- Tolerance №1: 0.3.

A section of the resulting DEM is presented in Figure 12 while the distribution of the primary (bathymetric) input data is shown in Figure 13. The channel morphology model was generated by a many magnitudes higher number of points by the currently proposed method compared to the former cross-sectional based surveys, therefore, presumably, a bathymetry of much higher accuracy is available for end-users. The red lines in Figure 13, composed of individual measurement points, indicate the measurements trajectories, while bathymetric points to the left and right of the trajectory are positioned in a cross-sectional pattern.

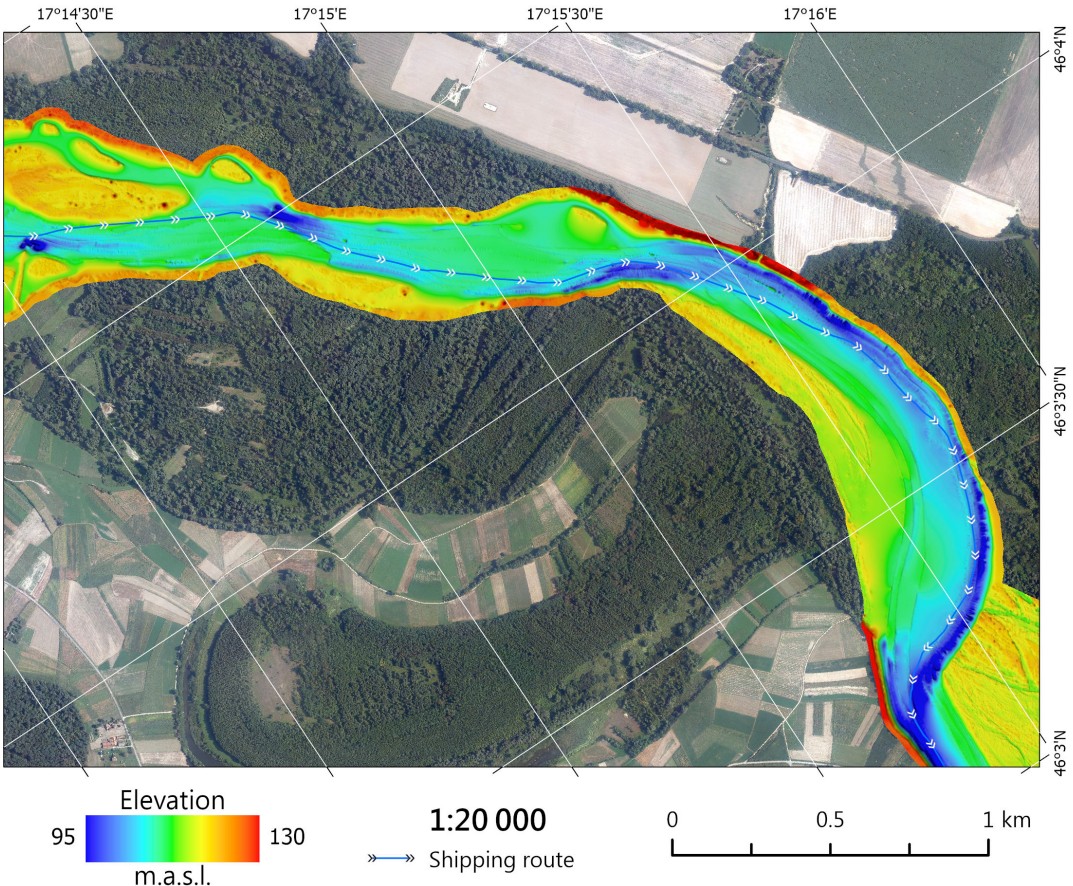

**Figure 12.** A section of the resulting DEM, near Heresznye (Hungary, on the northern side of the river) and Ferdinandovac (Croatia, on the southern side of the river). The river flows from left to right (west to east). The distribution of the bathymetric input data are presented on Figure 13. See the text above for detailed description of input datasets and ANUDEM settings. The resolution of the resulting raster is $1 \times 1$ meter.

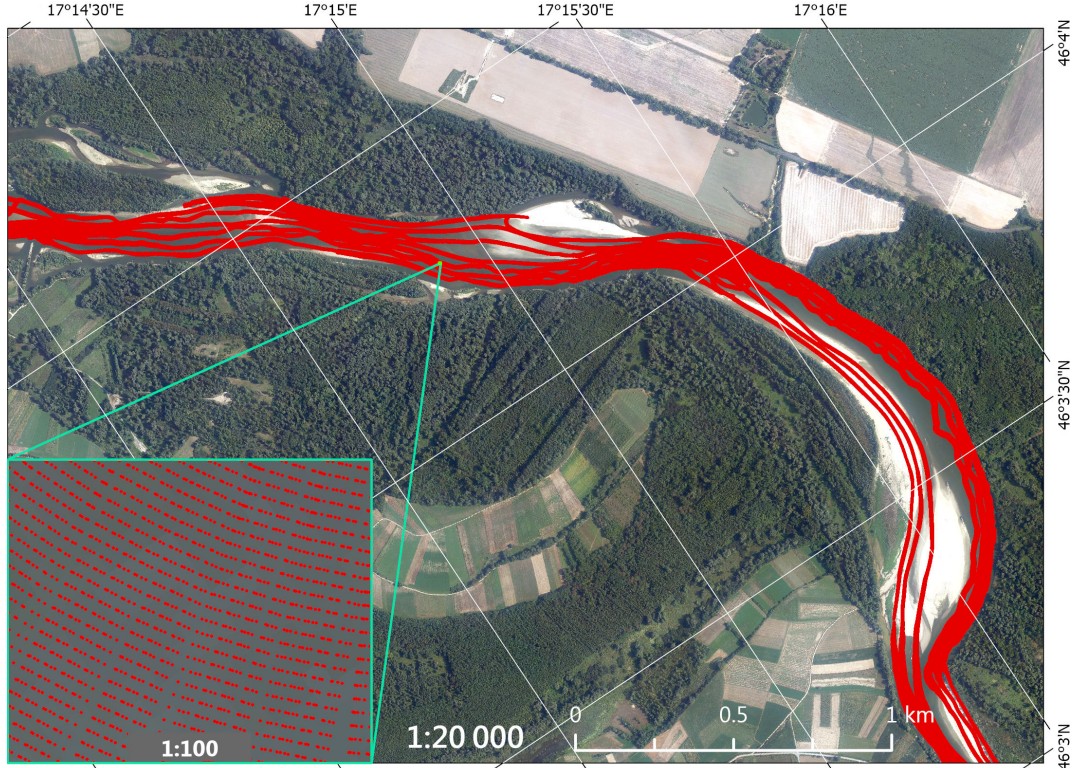

**Figure 13.** River bathymetry of the channel at the village of Heresznye (same location as in Figure 12). Each seemingly solid line on the main map (1:20 000) consists of thousands of cross-section-like points based on the individual readings of the interferometric transducer (see the inlet map, 1:100, at the left side).

To validate our bathymetric results a preliminary accuracy test was performed with a Ohmex (Sway, United Kingdom) SonarM8™ single-beam sonar in July, 2018 at the village of Drávaszabolcs, Hungary (78+000 rkm). These readings revealed a maximum deviance of ±0.2 m between the single-beam and interferometric sonar outputs. Additional differences were also caused by the intrinsic errors of the sonar used for validation and the moving underwater objects that were captured by one method and were rejected by the other. However, verification, accomplished in an environment of known geometry and using a non-sonar technology (e.g., an extensive GNSS survey), will be indispensable for an objective assessment of recreational-grade interferometric sonar performance. Ideal environmental conditions for a verification procedure of this sort would be still and clear water bodies with a thin layer of sediments without any floating objects.

## 4. Discussion

Our findings revealed that our proposed method was applicable for the bathymetric survey of the 146 km-long Hungarian reach of the Drava. Our method markedly shortened the actual survey time, compared to the cross-section based approach, which eventually took several months. The newly elaborated and herein described method reduced the measurement campaign to a period of 20 days. Data processing in the case of the cross sectional method required a plethora of post processing work and a large number of semi-automated interpolation procedures whilst the data density remained low. With the advent of the recreational-grade sonars and the methods are presented here the processing algorithm became automated in 95% and it does not demand any auxiliary datasets besides the banklines and the outlines of the in-channel islands and the elevation data of these datasets. In addition, the new interferometric fish-finder-based survey system is feasible for the efficient and low budget surveys of long sections of relatively shallow rivers.

Filtering and correction of erroneous elevation data became automated therefore a reliable terrain model was obtained. Compared to the cross-sectional data acquisition method data of 10 orders of magnitude higher number of measured points is available for the representation of channel bathymetry and morphology. Moreover, visualization methods based on sidescan and DownScan™ also provided additional data on channel morphology.

Due to the semi-automated interpretation of sidescan and DownScan™ images, the location and dimensions of structural objects, potentially hazardous for riverine navigation, were identified. During the current study, we have developed a bathymetry map, which includes the detected hazardous objects in the channel, therefore suitable for the delineation of navigational routes. The high-resolution bathymetry of the Drava channel bed also supports sustainable tourism goals of the area and facilitates international navigation on the Drava River. For the visualization of the navigational route, a complex mapping algorithm with standardized resolution, coloring and legend has been developed which provides uniform outputs in an automated way. The obtained navigational map provides indispensable spatial data for all stakeholders in the hydrological sector. The model also provides exact geographical locations where structural and channel modification interventions are needed for the lateral and vertical protection of the riverbed.

Due to the high resolution of the developed elevation model, the morphological evolution of the channel is detectable while the model provides data of augmented precision for hydrological models of multiple dimensions. This is especially important for the Drava due to the relatively non-anthropogenic nature of the channel and its immediate riparian zone in which significant floods may substantially reshape the riverbed. Hence, the newly developed elevation model may be used for the prediction of future channel evolution and the mitigation of economic losses and potential hazards to human lives in the broader urbanized floodplain.

## 5. Conclusions

We have developed a new survey algorithm using a recreational-grade, low-budget sonar for the survey of channel bathymetry and to mark the navigational and riverine transportation route on the Drava River. To process obtained field data, a novel software was developed which transforms raw data into common GIS data formats. The interferometric (3D) bathymetric data, the sidescan and DownScan™ imagery jointly provided the foundation of a new survey methodology which was applied during the current research. With the acquired survey derivatives, data of higher spatial accuracy and precision could be supplied for hydrodynamic and sediment transport modeling and for the navigation on rivers of rapidly changing channel morphology. The developed hardware system is flexible and freely expandable, and provides room for further development, including the smoothing of the butterfly pattern in the processing algorithm. To improve the spatial accuracy of the bathymetric data for the comprehensive spatial representation of the channel the present sonar system will be advanced with a camera array, a radar and a mobile terrestrial laser scanner.

**Author Contributions:** Conceptualization, Ákos Halmai and Ervin Pirkhoffer; methodology, Ákos Halmai and Ervin Pirkhoffer; software, Ákos Halmai; visualization, Ákos Halmai; data curation, Alexandra Gradwohl–Valkay, Johanna Ficsor and Kinga Kiss; funding acquisition, Ervin Pirkhoffer; project administration, Ervin Pirkhoffer; supervision, Ervin Pirkhoffer; validation, Ákos Halmai; writing—original draft, Ákos Halmai; writing—review and editing, Szabolcs Czigány, Dénes Lóczy and Zoltán Árpád Liptay. All authors have read and agreed to the published version of the manuscript.

**Funding:** This research was funded by the Higher Education Institutional Excellence Program of Ministry of Human Capacities (Hungary), grant number "20765-3/2018/FEKUTSTRAT" at University of Pécs; and the Hungarian Scientific Research Fund (project GINOP-2.3.2-15-2016-00055).

**Acknowledgments:** The authors are grateful to the South-Transdanubian Water Management Directorate for providing various datasets (LiDAR, orthoimagery, geometry of artificial objects) and their devoted assistance during the measurement campaigns.

**Conflicts of Interest:** The authors declare no conflict of interest.

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
