# Peer review of "Applicability of a Recreational-Grade Interferometric Sonar for the Bathymetric Survey and Monitoring of the Drava River"

_ijgi, doi:10.3390/ijgi9030149_

Round 1

Reviewer 1 Report

This study is very important and will gain attention at a time when world's water needs to be mapped by 2030.

Few things to note for improvements:

Prune well for minor typos - e.g., Line 625. Line 302

You mentioned in Line 618 that a navigational map was developed, but there is nowhere in the paper to visualize this?

It will be very helpful if the challenges/limitations of this system is discussed. You said in Line 643 that the system provides room for further development, what sort of development?

There are two many 'very' short paragraphs, consider condensing some of your statements. Also, stick to one paragraphing. 

In Line 54 - 56, it challenge(s) of your bathymetry system is highlighted, consider expanding this paragraph by explaining how the properties mentioned would be challenging, and what other bathymetry systems (not limited by these challenges) could offer solutions.

Given the sensitive nature of this owrk (i.e. bathymetry for navigation safety), it will be useful if some validation exercise take place to compare this system with existing ones - to reveal the strengths and its robustness.

Author Response

Prune well for minor typos - e.g., Line 625. Line 302.

Thank you for pointing them out, we have corrected both typos and many others.

You mentioned in Line 618 that a navigational map was developed, but there is nowhere in the paper to visualize this?

Thank you for pointing out this flaw. We have reworded the sentence as follows: “During the current study we have developed a bathymetry map, which includes the detected hazardous objects in the channel, therefore suitable for the delineation of navigational routes.”

It will be very helpful if the challenges/limitations of this system is discussed. You said in Line 643 that the system provides room for further development, what sort of development?

We have reworded the last two sentences of the Conclusions chapter as follows: “The developed hardware system is flexible and freely expandable, and provides room for further development, including the smoothing of the butterfly pattern in the processing algorithm. To improve the spatial accuracy of the bathymetric data for the comprehensive spatial representation of the channel the present sonar system will be advanced with a camera array, a radar and a mobile terrestrial laser scanner.”

There are two many ‘very’ short paragraphs, consider condensing some of your statements. Also, stick to one paragraphing. 

We have merged these short paragraphs, it was also suggested by Reviewer №2.

In Line 54 - 56, it challenge(s) of your bathymetry system is highlighted, consider expanding this paragraph by explaining how the properties mentioned would be challenging, and what other bathymetry systems (not limited by these challenges) could offer solutions.

To explain the undesired properties of traditional sonars, the following two sentences have been inserted into the manuscript: “Small rivers, however present specific challenges for bathymetric surveys, as large devices cannot be mounted on shallow-draft vessels and traditional sonars are not designed to survive frequent impacts with large floating or non-visible subsurface objects. From the viewpoint of riverine traffic, small water courses have low priority; therefore no significant financial resources are available for their survey.” 

Given the sensitive nature of this owrk (i.e. bathymetry for navigation safety), it will be useful if some validation exercise take place to compare this system with existing ones - to reveal the strengths and its robustness.

Thank you for the comment; we have added a new paragraph at the end of the Results. The paragraph reads like this: “To validate our bathymetric results a preliminary accuracy test was performed with an Ohmex SonarM8™ single-beam sonar in July, 2018 at the village of Drávaszabolcs, Hungary (78 + 000 rkm). These readings revealed a maximum deviance of ± 0.2 m between the single-beam and interferometric sonar outputs. Additional differences were also caused by the intrinsic errors of the sonar used for validation and the moving underwater objects that were captured by one method and were rejected by the other. However, verification, accomplished in an environment of known geometry and using a non-sonar technology (e.g. an extensive GNSS survey), will be indispensable for an objective assessment of recreational-grade interferometric sonar performance. Ideal environmental conditions for a verification procedure of this sort would be still and clear water bodies with a thin layer of sediments without any floating objects.”

Thank you for reviewing our manuscript.

Reviewer 2 Report

This manuscript could be published after minor revision.

comments

Line 42-43. You could add some references like the following.

D. Williams, "Cubist-Inspired Deep Learning with Sonar for UXO Detection and Classification," Final Report, SERDP Project MR18-1444, January 2019.

B. Gips and D. Williams, "Through-the-Sensor Performance Estimation of the Mondrian Detection Algorithm in Sonar Imagery," Proceedings of IEEE OCEANS 2018, Charleston, South Carolina, October 2018.

Line 53. You could add the “environmental purposes” and “cultural heritage assessment surveys’” with references like the following.

Gournia, C.; Fakiris, E.; Geraga, M.; Williams, D.P.; Papatheodorou, G. Automatic Detection of Trawl-Marks in Sidescan Sonar Images through Spatial Domain Filtering, Employing Haar-Like Features and Morphological Operations. Geosciences 2019, 9, 214.

Ferentinos G., Fakiris E., Christodoulou D., Geraga M., Dimas X., Georgiou N., Kordella S., Papatheodorou G., Prevenios M., Sotiropoulos M., 2020. Optimal sidescan sonar and subbottom profiler surveying of ancient wrecks: The ‘Fiskardo’ wreck, Kefallinia Island, Ionian Sea. Journal of Archaeological Science. V. 113, Jan 2020, 105032.

Line 42-56. You could merge these small paragraphs.

Line 59. You could add some references like the following.

Wada, Masaaki & Yasui, Shigeya & Saville, Ramadhona & Hatanaka, Katsumori. (2014). The development of a remote fish finder system for set-net fishery. 10.1109/OCEANS.2014.7003174.

Line 57-81. You could merge these small paragraphs.

Line 82-102. The motivation paragraphs could also be merged.

Figure 5. This figure does not have to be that large.

Figure 13. Add more interpretation details.

Discussion. Please reform the paragraphs and discuss the results
with regard to the questions of Lines.83-88.

Author Response

Line 42-43. You could add some references like the following…

We have added the following to references to the text:

  1. Gips, B.; Williams, D. P. Through-the-sensor performance estimation of the Mondrian detection algorithm in sonar imagery. 2018, Charleston, SC, U.S.A., 22 – 25 Oct. 2018, DOI: 10.1109/OCEANS.2018.8604710
  2. Williams, D. Cubist-Inspired Deep Learning with Sonar for UXO Detection and Classification. Final Report, SERDP Project MR18-1444, January 2019.

Line 53. You could add the “environmental purposes” and “cultural heritage assessment surveys” with references like the following…

We have inserted both references as requested by the Reviewer:

  1. Gournia, C.; Fakiris, E.; Geraga, M.; Williams, D.P.; Papatheodorou, G. Automatic Detection of Trawl-Marks in Sidescan Sonar Images through Spatial Domain Filtering, Employing Haar-Like Features and Morphological Operations. Geosciences 2019, 9, 214. DOI: 10.3390/geosciences9050214
  2. Ferentinos, G.; Fakiris, E.; Christodoulou, D.; Geraga, M.; Dimas, X.; Georgiou, N.; Kordella, S.; Papatheodorou, G.; Prevenios, M.; Sotiropoulos, M. Optimal sidescan sonar and subbottom profiler surveying of ancient wrecks: The ‘Fiskardo’ wreck, Kefallinia Island, Ionian Sea. Journal of Archaeological Science 2020, 113, pp. 1 – 11. DOI: 10.1016/j.jas.2019.105032

Line 42-56. You could merge these small paragraphs.

We have merged the three paragraphs; indeed, it makes more sense like this.

Line 59. You could add some references like the following…

The recommended reference has been added to the text:

  1. Wada, M.; Yasui, S.; Saville, R.; Hatanaka, K. The development of a remote fish finder system for set-net fishery. 2014, Oceans – St. John’s, St. John’s, NL, Canada, 14 – 19 Sept. 2014, DOI: 10.1109/OCEANS.2014.7003174

Line 57-81. You could merge these small paragraphs.

We have merged the paragraphs as requested.

Line 82-102. The motivation paragraphs could also be merged.

We have merged the two paragraphs below the research questions; however, we have left the enumerated research questions in their original form.

Figure 5. This figure does not have to be that large.

We have decreased the height of this figure.

Figure 13. Add more interpretation details.

The following two sentences have been inserted into the text: “The channel morphology model was generated by a many magnitudes higher number of points by the currently proposed method compared to the former cross-sectional based surveys, therefore, presumably, a bathymetry of much higher accuracy is available for end-users. The red lines in Figure 13, composed of individual measurement points, indicate the measurements trajectories, while bathymetric points to the left and right of the trajectory are positioned in a cross-sectional pattern.”

The figure caption has been reworded: “River bathymetry of the channel at the village of Heresznye (same location as in Figure 12). Each seemingly solid line on the main map (1 ∶ 20 000) consists of thousands of cross-section-like points based on the individual readings of the interferometric transducer (see the inlet map, 1 ∶ 100, at the left side).”

Discussion. Please reform the paragraphs and discuss the results with regard to the questions of Lines.83-88.

We have merged most of the paragraphs here and reworded the first paragraph of the Discussion with regard to the motivation questions as flows: “Our findings revealed that our proposed method was applicable for the bathymetric survey of the 146 km-long Hungarian reach of the Drava. Our method markedly shortened the actual survey time, compared to the cross-section based approach, which eventually took several months. The newly elaborated and herein described method reduced the measurement campaign to a period of 20 days. Data processing in the case of the cross sectional method required a plethora of post processing work and a large number of semi-automated interpolation procedures whilst the data density remained low. With the advent of the recreational-grade sonars and the methods are presented here the processing algorithm became automated in 95% and it does not demand any auxiliary datasets besides the banklines and the outlines of the in-channel islands and the elevation data of these datasets. In addition, the new interferometric fish-finder-based survey system is feasible for the efficient and low budget surveys of long sections of relatively shallow rivers.”

Thank you for reviewing our manuscript.